# Building molecular model series from heterogeneous CryoEM structures using Gaussian mixture models and deep neural networks
Muyuan Chen ⊠

Cryogenic electron microscopy (CryoEM) produces structures of macromolecules at near-atomic resolution. However, building molecular models with good stereochemical geometry from those structures can be challenging and time-consuming, especially when many structures are obtained from datasets with conformational heterogeneity. Here we present a model refinement protocol that automatically generates series of molecular models from CryoEM datasets, which describe the dynamics of the macromolecular system and have near-perfect geometry scores. This method makes it easier to interpret the movement of the protein complex from heterogeneity analysis and to compare the structural dynamics observed from CryoEM data with results from other experimental and simulation techniques.

Single particle cryogenic electron microscopy (CryoEM) has emerged as the leading technique for determining the structures of proteins and RNAs in recent years[1–3]. Instead of producing one static 3D structure per sample, CryoEM provides the opportunity to explore the dynamics of the macromolecular systems. The in silico classification techniques enable the determination of multiple states from a single dataset, revealing the assembly steps of a macromolecular complex[4,5], or functioning mechanisms of membrane channels[6,7]. With advanced machine learning methods, particles can be mapped to a latent space that describes the conformational landscape of the system and provides insights into the continuous movement of different domains of macromolecules[8–11].

As our capability of heterogeneity analysis expands, interpreting the conformational landscape of a macromolecular system and building molecular models for the many CryoEM maps produced by the analysis becomes a major bottleneck. For a single high-resolution structure, an atomic model can be built by refining homolog or predicted models against the map[12,13], or using the de novo modeling tools directly from the 3D reconstruction[14,15]. While existing tools can automatically produce reasonable starting models, it is often time-consuming to refine them and get final models that fit well into the maps and have good stereochemical geometry. In addition to running automated refinement software[16,17], the modeling process frequently involves multiple iterations of semi-manual model adjustment[18], as well as repeated submission to the PDB validation server[19] to achieve satisfying validation scores.

When a system involves continuous conformational changes, the modeling task becomes more challenging. To describe a movement trajectory, one strategy is to build one model at each distinct conformation and morph between the models by interpolation[20,21]. However, simple interpolation does not necessarily capture the transition between the conformational states, and the intermediate models along the morphing trajectories are not guaranteed to have the correct stereochemical geometry. An alternative approach is to incorporate the constraints and force fields of the molecular dynamics (MD) simulation during the structure variability analysis of the particles[22,23], but these methods are computationally expensive for large complexes. To model the transition between conformational states captured from the CryoEM data and describe the dynamics of macromolecular systems, new modeling approaches are needed to build an ensemble of molecular models from particles of different conformations, using the structural variability information captured by heterogeneity analysis techniques.

Previously, we showed the Gaussian mixture model (GMM)-based protein structure representation, combined with deep neural networks (DNN), can be used to extract structural variability information from CryoEM datasets[10,24]. Here, we present a computational method, similarly based on the GMM-DNN architecture, that builds a series of molecular models from a CryoEM dataset with structural variability. Each model in the ensemble would satisfy protein geometry constraints and fit to the reconstruction of particles from the same conformation at the target resolution.

Division of CryoEM and Bioimaging, SSRL, SLAC National Accelerator Laboratory, Stanford University, Menlo Park, CA, USA. ⊠e-mail: muyuanc@stanford.edu

## Results

### Deep neural network for the refinement of a single molecular model

The modeling process starts from an existing homolog or predicted model, where we place one Gaussian function at each non-hydrogen atom. Using a DNN, we first refine the GMM so it fits the consensus 3D reconstruction of all particles. Then, taking the existing result from the heterogeneity analysis of a CryoEM dataset, the algorithm produces a series of atomic models, each corresponding to a conformation from the same point of the latent space.

During the refinement, the map-model Fourier shell correlation (FSC) is used to guide the model fitting, and a cutoff frequency is set to avoid over-interpreting the high-resolution information in the map. In contrast to existing real-space refinement methods[16,17,25], the properties of the GMM allow the algorithm to optimize the coordinates of atoms in the real space, while evaluating map-model similarity in the Fourier space. In addition to the FSC, we also introduce stereochemical constraints of proteins and RNAs to the loss function during DNN training, which ensures that the model at every frame along the trajectory has valid geometry. To enable gradient-based optimization for DNNs, we re-implemented the empirical constraints, including Ramachandran plots and sidechain rotamer libraries[26], as differentiable forms. The DNN-based optimization makes it possible to converge to a globally favorable geometry faster without human interference.

To demonstrate the protocol, we start with a classical modeling task, where one atomic model is refined against one high-resolution CryoEM structure. In this example, we refine an existing model of TRPV1 (PDB-3J5R)[27], built from a lower resolution structure, and fit it to a higher resolution map (EMD-8117, 2.95 Å)[28]. This is performed through an automatic multi-step refinement process, using three DNNs that capture structural information of different resolutions. The process involves large-scale morphing, residue-wise adjustment, and finally full atom refinement that considers both the map-model agreement and the stereochemical constraints (Fig. 1). The resulting model shows better real space fitting, higher Q-score[29], and near-perfect PDB validation metrics (Fig. 2A–D).

To show the robustness of the method, we applied it to 25 recent CryoEM structures from the PDB/EMDB, including proteins and RNAs, and ranging from 2.5 to 5.7 Å resolution. Fully automated refinement with default parameters yielded better geometry scores for every example, while the Q-score is largely unchanged (Figs. 2E and S5, 6). This indicates that our method improves the overall geometry of the molecular models without sacrificing the map-model agreement.

### Refinement of molecular model series using CryoEM data

Now the refinement protocol of single molecular models is established, we apply the method to refine a series of models from the previous heterogeneity analysis of CryoEM data. We again start from the TRPV1 example (EMPIAR-10059), from which the movement of Ankyrin repeat domains has been observed previously[11,24]. Starting from the heterogeneity analysis results, we follow a 1D trajectory in the conformation space, group particles by regular intervals, and generate 3D reconstruction from each group of particles. Then, given the positions of the particle groups in the conformational space as input, DNNs are trained to output molecular models that match the corresponding maps of the particle groups, resulting in a series of models that describe the conformational change captured by the heterogeneity analysis. In addition to the map-model similarity, the stereochemical constraints are also considered during the DNN training, so the output model from any conformational point would have minimal clashing and good geometry (Fig. S1 and Supplementary video 1).

Finally, we use the spliceosome dataset (EMPIAR-10180) as an example to demonstrate the capability of the protocol[30]. The spliceosome is a complex of more than 100,000 atoms, including both protein and RNA, and the large-scale structural flexibility of the system has been well documented[9,10,31,32]. Using our method, we build molecular models that describe the continuous movement within the system, which match the results of existing heterogeneity analysis. Moreover, each snapshot model along each movement trajectory has few atomic collisions and near-perfect geometry score (Fig. 3 and Supplementary video 2).

## Discussion

In sum, for single model refinement tasks, our atomic model refinement protocol produces better models than results from existing methods, according to the PDB validation metrics. While a perfect geometry score does not necessarily guarantee a good model, the flexible nature of the DNN makes it straightforward to adopt any new validation criteria in the future. The GMM-based architecture also connects the modeling step to the heterogeneity analysis of CryoEM datasets seamlessly, making it possible to build a series of models that describe the structural dynamics of the macromolecular system. The entire process is performed automatically without any manual intervention. Compared to stacks of 3D reconstructions, the molecular model series makes it easier to interpret the dynamics of the protein complex and provides a convenient way to compare the structure flexibility information obtained by CryoEM with results from other techniques, including MD simulation and NMR[33–35].

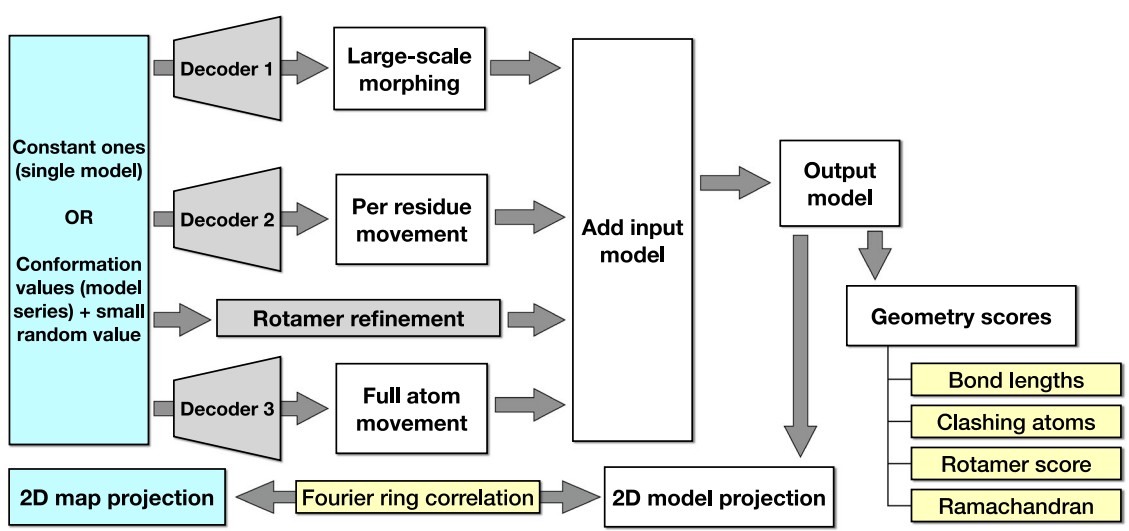

**Fig. 1 | Workflow diagram for the GMM-DNN-based molecular model refinement protocol.**

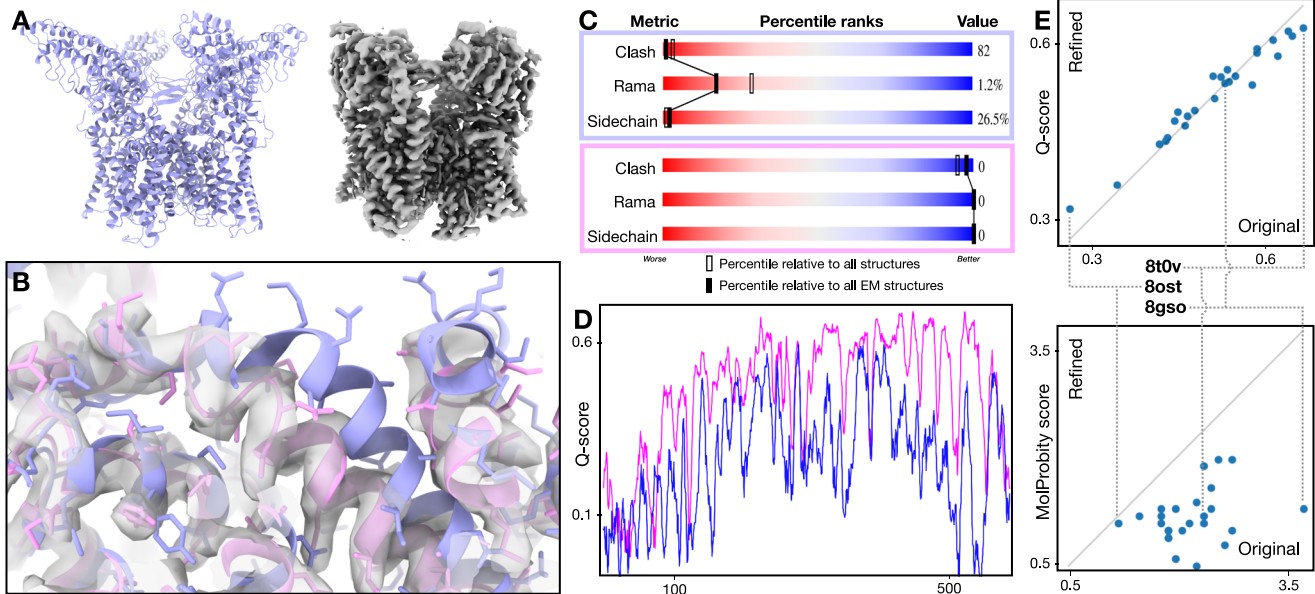

**Fig. 2 | Single model refinement. A** Input model (PDB-3J5R) and CryoEM map (EMD-8117) of TRPV1. **B** Zoomed-in view of map and models. Blue - input model; pink - GMM refined model. **C** PDB validation results of the input (top) and refined (bottom) model. **D** Q-score comparison of the input (blue) and refined (pink) model. **E** Comparison of the original and refined versions of 25 models from the PDB. Top - Q score; bottom - MolProbity score (lower score indicates better geometry).

## Methods

### GMM for atomic model representation

The basic concepts of GMMs and the architecture of the DNNs have been described in our previous publications. In this section, we briefly explain the basic concepts of the method, but it is recommended to refer to the previous papers for more details[10,24,36].

To represent the structure of proteins, we use a GMM, which is a sum of many Gaussian functions in real space, with each Gaussian function placed at one non-H atom. Each Gaussian function is represented by five parameters: amplitude, width, and the 3D center coordinates. Similar to our previous GMM-DNN-based heterogeneity analysis, to reduce computational resource consumption, the DNN is trained on batches of 2D projection images instead of 3D volumes. To compare projections of the GMM and projections of the map, we use the Fourier ring correlation (FRC) as the loss function. The FRC between the Fourier transform of two images is the average of the correlation coefficients over Fourier rings. Since each ring is independently normalized, the FRC is insensitive to filtering of images or the sharpening methods applied to the reconstructions.

### Basic stereochemical constraints

A number of stereochemical constraints were included for the model refinement. It is worth noting that here we only use static constraints that describe the quality of output model, but not any of the force fields from MD simulation. As such, the conformational changes captured by our modeling approach are only driven by the underlying CryoEM data.

We first consider the length and angle of the covalent bonds in the molecule. Initially, we take the mean and standard deviation (std) of each type of bond for every amino acid residue from AlphaFold[12]. However, when refining models with those parameters, we notice that there is an inconsistency between the AlphaFold values and the statistics used in the PDB validation server. For some types of bonds, smaller std values are used in the PDB validation, so outliers become more likely when refining with those constraints. To compensate for this, we gathered information from many PDB validation reports and corrected those values manually, so the statistics agreed with PDB validation.

In reports from the PDB validation server, bonds with lengths that are >5 std from the average bond length value of that type are considered

outliers and receive a penalty in the validation score. To enforce the ideal bond length and angle, during DNN training, we included both the likelihood that the current bond length and angle follow the Gaussian distribution of the target mean and std, as well as an additional penalty for outliers. Here we set the threshold for outliers to be 4.5 std by default, slightly tighter than the PDB validation server requirement, to overcome the potential error caused by the PDB file format, as well as any remaining discrepancy between the ideal bond values used by us and the validation server.

In addition to the bond length and angles, we also compute dihedral angles from the model. Here, we first apply the constraints for the planar dihedral angles, which enforce sets of atoms to always sit on the same plane. Keeping consistent with the PDB validation server, the threshold of the maximum allowed peptide bonds dihedral angle is set to be 30°, and the threshold for other planar bonds, including amino acid backbone and some planar sidechains, to be 10°.

During the DNN training, all bonds and angles are directly computed from the GMM, using the center of Gaussian functions as the coordinates of the atom. To speed up the process, before the refinement, we pre-compile a list of atom pairs that form bonds, as well as the ideal bond length and std for each bond on the bond type. During the refinement, for each batch, the program evaluates the length of all the bonds in the list and compares them to the ideal values. Similarly, we pre-compile the 3-atom lists for bond angle and 4-atom lists for dihedral angle calculation, so the geometry can be quickly evaluated during DNN training.

### Ramachandran plot

Ramachandran plot describes the preferred geometry of peptide chains using dihedral angles along the protein backbone. With two angles per residue, it is essentially a 2D histogram with finite sampling (often at 1°). While it is straightforward to compute a Ramachandran score from any protein backbone, it is challenging to use the score as a part of the loss function during the DNN training, because the analog gradient, which is required by most DNN optimizers[37], cannot be computed from the discrete histogram. To overcome this, we need to convert the Ramachandran plot into a differentiable form. Here, similar to our 3D GMM-based protein density representation, we fit 2D GMMs to the

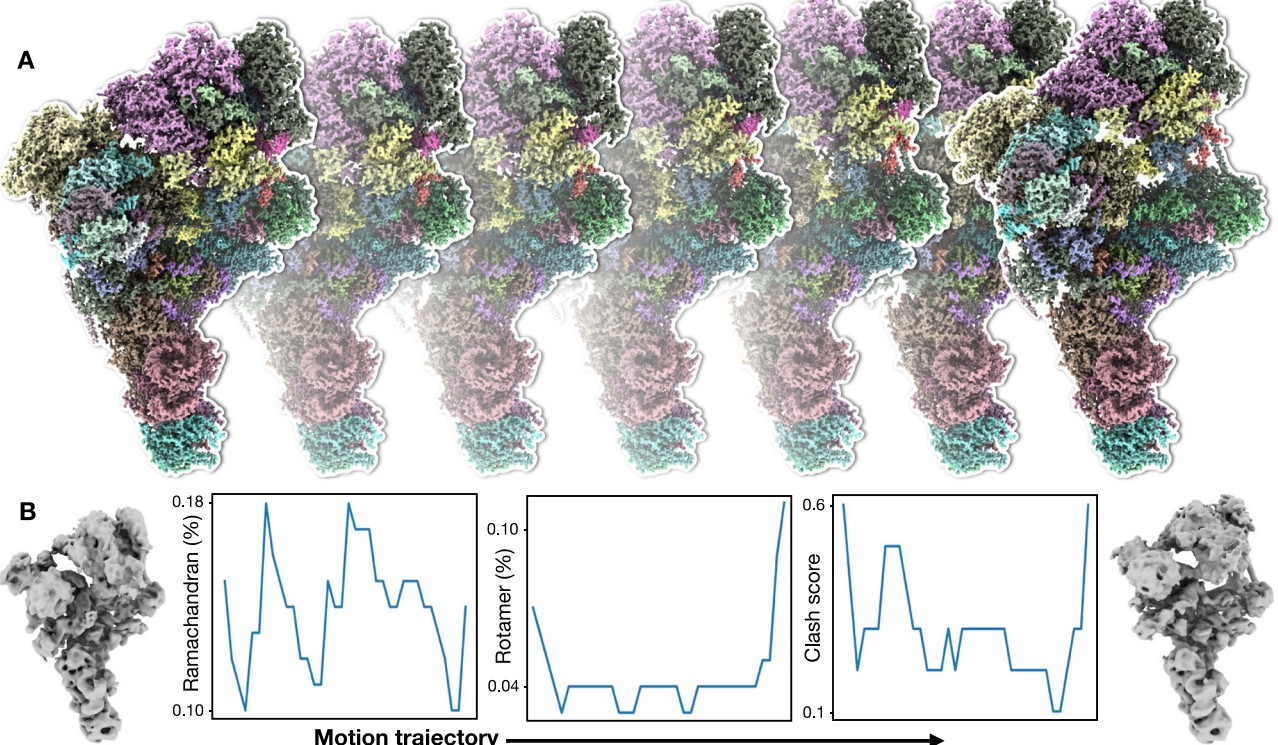

**Fig. 3 | Model series refinement. A** Snapshots of GMM refined molecular model series along one continuous motion trajectory of spliceosome from dataset EMPIAR-10180. **B** Left and right panels - 3D reconstruction of the first and last frame of the motion; middle three panels - Ramachandran, rotamer outlier, and clash score for models along the motion trajectory.

Ramachandran plots, turning the discrete histogram into a continuous function (Fig. S2).

For the GMM representation of Ramachandran plots, we use the six 2D histograms for the different residue types defined in Phenix[38]: General, Gly, trans-Pro, cis-Pro, pre-Pro, and Ile. For each type, 2500 Gaussian functions are used to fit the histogram. Because the threshold that determines an outlier in the Ramachandran plot in PDB validation is very small (0.0005, while the highest value in the histogram is 1), even when the GMM fitting is good, there can still be many Ramachandran angles that are considered acceptable by the GMM representation but are outliers in the histograms. Since Ramachandran outlier count is the main metric used by the PDB validation to evaluate the protein backbone, here we fit the logarithm of Ramachandran score ($\ln(R + C)$, where $R$ is the original Ramachandran score, and $C$ is a small constant set to $e^{-10}$). We also introduce additional penalties along the outlier boundaries of the plot during the GMM fitting, ensuring all Ramachandran angles that are considered outliers from the discrete histogram are also considered outliers in the GMM representation. Similar to the GMM representation of protein structures, the fitting of 2D GMM for Ramachandran plots is also done by densely connected neural networks. The RMSD of the GMM fitting is smaller than 0.001, and virtually all outliers in the histograms are outliers in the GMM representation.

With the GMM-based differentiable Ramachandran plot, we can then compute the Ramachandran score from the dihedral angles of any given model. Similar to the metrics used in PDB validation, here we only include the percentage of "allowed" and "outlier" residues as a part of the loss function during the DNN training. With the smooth GMM representation of the Ramachandran plot, the optimizer can compute the gradient of the score with respect to the atom coordinates automatically for any atomic model.

**Sidechain rotamers**

Similar to the Ramachandran score, the preferred sidechain rotamers are also defined as discrete histograms of the Chi angles of the amino acid sidechain. So similarly, we fit GMMs to those histograms in order to have a differentiable loss function (Fig. S3). Here, we use the rotamer library from Phenix. The number of Chi angles varies from 0 (Gly) to 4 (Arg, Lys) depending on the type of amino acid, so the dimension of the GMMs also changes accordingly. For all the amino acid types, the preferred rotamer histogram consists of a few Gaussian-like peaks on a near-zero background, making GMM fitting relatively simple. For sidechains with 1, 2, 3, and 4 Chi angles, we use 4, 25, 64, and 160 Gaussian functions, respectively, to fit the rotamer histograms. Since there are only a few peaks in the rotamer space, to fit the GMM, we simply place the center of Gaussian functions at those peaks and optimize the GMM parameters locally. Similar to the Ramachandran fitting, to increase the weights of small value outliers, here we fit the logarithm of Rotamer scores instead of the original values. After generating a GMM for each type of amino acid side chain (excluding Gly), the gradient of the rotamer score with respect to the coordinates of atoms can also be computed by automatic differentiation. Similar to the bond length and angle constraints, for sidechain rotamers, we consider both the likelihood that the Chi angles follow the Gaussian distribution and the additional penalties for rotamer outliers.

In addition to including the rotamer constraints as a part of the model refinement, we also offer an option to fully rebuild the rotamers from a given CryoEM map. To do this, we enumerate all peaks in the GMM of the sidechain rotamer for each residue and pick a rotamer with the best map model similarity. Afterward, the Chi angles of all rotamers are refined locally to maximize both the map and the geometry score. This functionality is particularly useful when fitting models built from low-resolution maps, where the sidechain rotamers are unreliable, into high-resolution maps with clear sidechain densities. In those situations, it can be difficult to converge to a different rotamer that agrees with the map from the starting point, because good rotamers only form isolated peaks in the space of Chi angles.

**Avoiding clashing atoms**

Clashing of atoms in a molecular model is detected when the distance between two unlinked atoms is smaller than the sum of their van der Waals

(VdW) radius, minus a certain threshold value (0.4 Å in PDB validation). Initially, we used the VdW radius of each atom in each amino acid type from AlphaFold. However, like in the case of bond length/angle, there turned out to be a discrepancy between the VdW radius from AlphaFold and those used in the PDB validation server. It is also notable that the VdW radius values used in the PDB validation server are also different from those used in Phenix, which further complicates the process. To obtain the precise VdW radius values for all atom types from the PDB validation server, we upload multiple synthetic atomic models to the server, which include clashing atom pairs of all atom types from every amino acid residue. The VdW radius of the different atom types is then directly parsed from the validation report and used in the refinement protocol.

To calculate the atom clashing score during model refinement, for each atom in the model, we track a number of neighboring atoms (default 128) that it might clash with, since computing an all-vs-all contact map at every iteration would be too expensive. The neighboring atoms are selected using the KD-Tree algorithm at the beginning of the refinement, and any atom pairs that are directly bonded or indirectly connected within 3 bonds are excluded. The neighbor list is updated every 100 iterations of refinement, in case faraway atoms start to form contact as the models morph during refinement.

Another major issue when calculating clashing atoms is the introduction of H atoms. As described earlier, H atoms are not included in the GMM, since they are generally invisible in CryoEM maps before reaching true atomic resolutions[39]. However, since H atoms are used in the PDB validation server for clashing score calculation, we have to add them back to the model to optimize for clashing. The H positions are based on the coordinates of the non-H atoms of the corresponding residue, and the exact relative positions of H atoms in all amino acid types are adapted from Phenix. To simplify the process, the DNN outputs models of only non-H atoms, and the H atoms are added to that model as a post-processing step during training.

There are two additional points of concern when computing the clash between H and other atoms. First, when a potential H-bond can form between two atoms, we increase the threshold that defines clashing between the two atoms by 0.4 Å, i.e., the atoms can stay closer without being considered clashing. Here, H-bonds are simply defined as a pair of H and O/N atoms that are not covalently bonded, and the angle of H-bonds is not considered.

Second, while in theory some of the H atoms, such as those in -CH3 or -OH, have an extra degree of freedom and can rotate around the bond, strangely, it turned out that the PDB validation server does not take this into consideration when computing clash scores even though the validation report claims that the H atoms are optimized. When the uploaded model includes H atoms, the validation server will remove all H atoms before adding the H atoms back at their default zero-degree torsion angles. Therefore, using optimized non-zero torsion angles for H atoms will lead to higher clash scores from the validation server, even if those clashes can be resolved by the rotation of H atoms. So, we forbid H rotation during the refinement and only place H atoms at the zero-degree conformation defined in the PDB validation server.

## RNA score

In the PDB validation, the RNA score is defined as the likelihood that the backbone of each RNA base belongs to one of the 46 suites[40]. Seven dihedral angles are computed from the backbone of each RNA base, and the suite score is calculated from the distance from the 7-angle vector of that base to the nearest pre-defined suite cluster center in the 7D space. Compared to the Ramachandran plot or the sidechain rotamers, the computation of the RNA score from the validation report is less intuitive. Specifically, RNA bases that do not belong to any suite (score smaller than 0.001 for any of the 46 clusters) are ignored in the final score, which is an average of the score from all bases that have an assigned suite. As a result, having outliers in RNA backbone geometry does not actually lead to a lower score, and each

non-outlier RNA base needs to get as close to the corresponding cluster center as possible to achieve a higher score.

Although the metric used in the validation does not consider outliers, during the refinement, we still assign each RNA base to the nearest suite center in the 7D space, without skipping bases with low scores. As such, an outlier RNA base with a near-zero score would still contribute to the overall RNA score. This prevents the DNN from converging to obviously incorrect high-score solutions, such as having only one base with a perfect score and everyone else an outlier.

## DNN-based single model refinement

Four-layer densely connected neural networks are used for the model refinement. The network structure is the same as the decoders used in heterogeneity analysis[10], which takes a conformation input and outputs the GMM parameters. Three neural networks are used for the refinement: one for the initial large-scale morphing, one for residue/base-wise refinement, and another for the full atom refinement. The outputs of the three neural networks are added together, all on top of the GMM built from the input model, so they can collaborate through a boosting-like mechanism. Starting from the long-range morphing, each network can refine the finer details of the model, target higher resolution, and consider more stereochemical constraints. The hierarchical design of the protocol ensures refinement can capture both the large-scale conformation change and the subtle geometry improvement, compared to the existing model refinement tool like Phenix (Table 1, and Figs. S4, 5).

Similar to our GMM-DNN-based heterogeneity analysis, to reduce resource consumption, the DNN is trained on batches of 2D projection images instead of 3D volumes. Since there is only one reference map in the single model refinement, the input to the DNN is always a 4D vector of constant ones ([1,1,1,1]). For each projection image in the batch, the DNN generates one GMM that includes all non-H atoms in the model. The GMM is projected to 2D, and FRC between the GMM projection and the projection of 3D volume is computed, which is used as the score for map-model similarity. From the GMM, various geometry scores can also be calculated, including the deviation of bond length and angle, the Ramachandran and rotamer scores, as well as penalties for clashing atoms.

For the model refinement, one major difficulty is to balance the weight between the score of map-model similarity and the score of the model geometry. To address this, an automated method is used to determine the weighting factor from the data itself. We start by running two iterations of the refinement that minimize the map-model similarity score without the model geometry constraints. From the refinement, we calculate the ratio between the increase of the map-model similarity score and decrease of the geometry score over the two iterations. This ratio quantifies how much model geometry quality is sacrificed to achieve the improvement of map fitting. Then, we set the weight of the geometry score to the reciprocal of that ratio and re-initialize the DNNs to restart the training. As a result, in the new iterations, the map-model similarity score and the geometry score will be roughly balanced, i.e., the geometry score changes at a similar scale as the map-model similarity during the training.

Generally, the refinement process includes five steps (Fig. 1). First, we pre-compile the geometry information from the model, so that all the geometry scores can be directly computed later as a part of the DNN training process. At this step, the input model file is parsed, and the information of all the bonds, angles, and dihedral angles, as well as their ideal values, are converted into matrix forms so the corresponding geometry scores can be calculated during the refinement.

Second, we train the first DNN to deal with the potential large-scale domain movement between the input model and the given CryoEM map. To model this, we use the hierarchical GMM architecture we previously developed[24], and divide the full atomic model into a small number of patches (64 by default). To maximize the map-model similarity, the DNN is trained to output the transform of each patch, which is then applied to the full GMM and morphs the atomic model. Here, the patches are divided automatically

**Table 1 | Comparison of refinement methods on 25 examples from PDB. (O, G, P for original, GMM, and Phenix refinement)**

| #PDB | #EMDB | Q-score (equivalent resolution) | | | Ramachandran outlier | | | Rotamer outlier | | | Clash score | | |
|------|-------|----------|------|--------|---|---|---|---|---|---|---|---|---|
| | | Original | GMM | Phenix | O | G | P | O | G | P | O | G | P |
| 8tjv | 41312 | 0.55 (3.16) | 0.55 (3.18) | 0.55 (3.15) | 0.00 | 0.00 | 0.00 | 0.0 | 0.0 | 0.0 | 10.5 | 1.3 | 10.7 |
| 8xzb | 38792 | 0.51 (3.37) | 0.54 (3.18) | 0.52 (3.30) | 0.51 | 0.00 | 0.13 | 4.5 | 0.1 | 0.1 | 19.7 | 0.0 | 9.2 |
| 8k3j | 36853 | 0.53 (3.26) | 0.53 (3.25) | 0.54 (3.22) | 0.17 | 0.00 | 0.17 | 0.3 | 0.0 | 0.3 | 8.5 | 2.1 | 8.8 |
| 9enf | 19833 | 0.43 (3.87) | 0.43 (3.83) | 0.44 (3.79) | 0.00 | 0.00 | 0.00 | 0.0 | 0.0 | 0.0 | 15.4 | 1.5 | 18.3 |
| 9c49 | 45178 | 0.59 (2.97) | 0.58 (2.98) | 0.59 (2.94) | 0.52 | 1.04 | 0.52 | 1.5 | 0.0 | 0.0 | 18.0 | 1.0 | 22.4 |
| 8xqa | 38571 | 0.42 (3.94) | 0.43 (3.87) | 0.42 (3.92) | 0.00 | 0.00 | 0.00 | 6.6 | 0.0 | 1.2 | 5.1 | 0.4 | 15.1 |
| 7yg9 | 33812 | 0.58 (2.98) | 0.59 (2.95) | 0.61 (2.84) | 0.00 | 0.00 | 0.00 | 0.0 | 0.0 | 0.0 | 13.8 | 1.4 | 21.8 |
| 8ost | 17164 | 0.26 (5.31) | 0.32 (4.75) | 0.32 (4.74) | 0.00 | 0.00 | 0.00 | 0.3 | 0.0 | 0.8 | 5.4 | 1.6 | 8.8 |
| 8t0q | 40945 | 0.61 (2.84) | 0.61 (2.87) | 0.61 (2.86) | 0.32 | 0.00 | 0.13 | 3.7 | 0.0 | 0.6 | 35.5 | 0.5 | 36.1 |
| 8wly | 37635 | 0.47 (3.63) | 0.48 (3.56) | 0.49 (3.51) | 0.11 | 0.00 | 0.11 | 1.1 | 0.1 | 0.3 | 11.7 | 1.0 | 12.3 |
| 8xse | 38617 | 0.62 (2.81) | 0.58 (3.01) | 0.62 (2.80) | 0.09 | 0.57 | 0.19 | 5.4 | 1.4 | 0.0 | 6.5 | 1.0 | 6.0 |
| 8unh | 42402 | 0.54 (3.22) | 0.54 (3.23) | 0.54 (3.22) | 0.24 | 0.18 | 0.18 | 0.5 | 0.1 | 0.1 | 46.8 | 1.5 | 50.5 |
| 8xqp | 38584 | 0.43 (3.85) | 0.44 (3.80) | 0.43 (3.86) | 0.00 | 0.00 | 0.00 | 0.3 | 0.0 | 0.5 | 14.4 | 1.8 | 14.0 |
| 8xqr | 38586 | 0.46 (3.66) | 0.46 (3.67) | 0.46 (3.64) | 0.00 | 0.19 | 0.10 | 0.0 | 0.0 | 0.9 | 13.5 | 0.8 | 13.8 |
| 8jtr | 36650 | 0.48 (3.55) | 0.48 (3.51) | 0.50 (3.45) | 0.48 | 0.29 | 0.48 | 0.1 | 0.1 | 0.4 | 7.8 | 0.4 | 11.6 |
| 8gso | 34237 | 0.52 (3.29) | 0.54 (3.19) | 0.55 (3.14) | 6.70 | 0.46 | 0.53 | 41.5 | 0.0 | 0.6 | 20.4 | 0.5 | 14.2 |
| 8t0v | 40947 | 0.67 (2.61) | 0.63 (2.78) | 0.65 (2.68) | 0.00 | 0.00 | 0.00 | 4.2 | 0.0 | 0.2 | 24.7 | 2.3 | 6.9 |
| 8k11 | 36783 | 0.54 (3.23) | 0.56 (3.12) | 0.58 (3.02) | 0.00 | 0.00 | 0.00 | 0.4 | 0.0 | 0.4 | 11.4 | 2.6 | 9.3 |
| 8xom | 38534 | 0.44 (3.76) | 0.47 (3.61) | 0.44 (3.81) | 0.42 | 0.00 | 0.00 | 1.1 | 0.1 | 0.4 | 33.8 | 13.8 | 18.3 |
| 8wyc | 37923 | 0.65 (2.69) | 0.61 (2.84) | 0.64 (2.74) | 0.00 | 0.11 | 0.00 | 6.7 | 0.3 | 0.4 | 38.7 | 0.6 | 12.2 |
| 8k7t | 36941 | 0.45 (3.73) | 0.48 (3.53) | 0.49 (3.47) | 0.37 | 0.28 | 0.37 | 0.6 | 0.0 | 0.4 | 12.1 | 2.5 | 18.3 |
| 8k03 | 36756 | 0.51 (3.36) | 0.51 (3.39) | 0.50 (3.42) | 0.00 | 0.17 | 0.00 | 0.0 | 0.0 | 0.4 | 13.8 | 0.1 | 11.8 |
| 8q74 | 18203 | 0.34 (4.50) | 0.36 (4.39) | 0.34 (4.50) | 0.90 | 0.38 | 0.77 | 0.1 | 0.0 | 0.6 | 21.0 | 2.2 | 18.3 |
| 9c57 | 45206 | 0.64 (2.72) | 0.62 (2.81) | 0.64 (2.73) | 0.38 | 0.25 | 0.36 | 2.2 | 0.2 | 0.6 | 11.9 | 0.9 | 10.7 |
| 8wx0 | 37896 | 0.58 (3.02) | 0.53 (3.26) | 0.55 (3.14) | 0.17 | 0.39 | 0.23 | 0.6 | 0.1 | 0.5 | 10.4 | 1.0 | 27.4 |

from the centers of all amino acid residues (or DNA/RNA bases) using K-means clustering, so atoms of the same residue/base are always in the same patch. This forces each residue/base to move as a group during the large-scale morphing and avoids introducing additional geometry errors within residues. The implementation of this step is essentially the same as the GMM-based heterogeneity analysis from CryoEM particles[10], except that projections of the given map are used instead of particle images, and the position of atoms is used in place of Gaussian functions generated from the density map.

In the third step, another DNN is introduced to adjust the model at the residue/base level. The second DNN is implemented using a similar approach as the first one, except each patch contains only atoms from one residue. At this step, the two DNNs are trained together, and their outputs are summarized so the second DNN can perform residue-level geometry adjustment on top of the large-scale domain movement learned by the first DNN in the previous step. Also, at this point, we start to introduce the basic geometry constraints, including bond length, angle, and clash score, into the loss function.

After the residue-wise refinement, we next optimize the sidechain rotamers. For high-resolution maps, as described above, we rebuild the rotamers for all residues first. Then the rotation of each sidechain Chi angles is optimized to better fit into the map while also keeping an acceptable rotamer score and not clashing with other atoms. Since this is only a local search, it is done without the DNN, by directly optimizing the Chi angles with the Adam optimizer[37].

Finally, we refine the full atom model with the third DNN, taking all the stereochemical constraints into consideration. This DNN generates the full GMM that includes all non-H atoms, and outputs from three DNNs, as well as the side chain torsion angles from the previous step, are summarized together to produce the final model. The final step can also run independently without an input map, to only refine the geometry of the model locally, without the constraints from structure data.

For each DNN after training, the information of the model is entirely stored within its weight matrices, and an input of a vector of ones will deterministically produce the refined model. So, to integrate the results of multiple DNNs in single model refinement, we do not need to save the weights of the networks, but only use the output DNN from the previous step as the baseline model for the training of the next DNN. That is, the input model is first subject to the domain motion refinement, then the output is used for the residue-wise refinement, and finally full atom refinement.

**Continuous model series refinement**

The input of the continuous model series refinement is a stack of 3D reconstructions, along with their corresponding assigned conformation values (i.e., location in the latent conformational space during the heterogeneity analysis). So, results from any continuous heterogeneity analysis protocol[9,11,32], in addition to our GMM-DNN-based one[10], can be used as the input for the modeling process. Same as the single model refinement, we make projections from the reconstructions, and use those 2D projections instead of 3D volumes as DNN training input to reduce the computational resource usage.

The procedure of model series refinement is generally the same as the single model refinement, but instead of a constant vector, here, for each projection image, the actual conformation value of the corresponding reconstruction is used as the input of the DNN. To ensure the continuity of the output models, a small random variable is added to the conformation input, similar to the variational autoencoder implementation[41]. Therefore, the output models can have continuous

movement even though the input reconstructions are only sampled at discrete positions in the conformational space.

Finally, to make sure the model series always has good geometry, we perform additional rounds of local geometry refinement using the input of uniformly distributed random variables along the trajectory in the latent space. As a result, the output model series would have near-perfect stereochemical scores even at the frames that are not sampled by the 3D volumes from the input.

Unlike the single model refinement, here the weights of the DNNs are in fact necessary to produce the model series from the input conformation values. Therefore, to integrate multiple networks, the DNNs need to be trained together. That is, during the large-scale morphing refinement, only the first DNN is trained, and its weights are stored for the subsequent training rounds. For the residue movement refinement, the first two DNNs are trained together and their output is summarized to produce the output model series, and during the full atom refinement, all three DNNs are trained together.

## Details on examples

For the TRPV1 example, we start from the PDB model (PDB-3J5R)[27], which was built from a lower resolution structure at a different conformation (EMD-5777, 4.2 Å). The model was first inspected and roughly fitted into the new map (EMD-8117, 2.95 Å)[28] using UCSF ChimeraX[42], before being used as the input for the refinement protocol. The target resolution for model refinement was set to 3 Å, and default parameters were used for the refinement. Since the map resolution is high enough, we also rebuilt all sidechain rotamers according to the density map during the refinement.

The continuous movement of TRPV1 was generated from the public dataset EMPIAR-10059, using the heterogeneity analysis previously described in ref. 24. The heterogeneous model series was built from a series of 3D reconstructions generated along the first eigenvector of the particle distribution in the conformation latent space, which describes a rotation motion of the ankyrin repeats domain. The output model from the previous single model refinement of TRPV1 is used as the input for the model series refinement. The target resolution of the model series refinement was set to 7 Å, and default parameters were used during the process (Supplementary video 1).

To model the continuous movement of the spliceosome, we use the dataset EMPIAR-10180[30], and heterogeneity analysis results previously described. The original model (PDB-5NRL) was used as the input for model refinement, and the 3D reconstruction from the neutral state of the heterogeneity analysis was used as the target map for the initial single model refinement[10]. A series of 3D reconstructions was generated along the first eigenvector in the conformation latent space, and the target resolution of the model series refinement is 15 Å (Supplementary video 2). Additionally, movement along a circular trajectory is also modeled (Supplementary video 3). Default parameters were used during the refinement, except that the batch size was set to 2 in order to fit the large complex in GPU memory.

To demonstrate the performance of the model refinement protocol in general cases, we picked 20 protein structures and 5 RNA structures from the PDB. To show that our method can improve structures with relatively poor geometry, we select models with validation scores at the lower 50% percentile using the advanced search function provided by PDBe[43]. To make sure the selected models were generated using the latest model refinement techniques after the PDB validation service was established, we sorted the PDB by deposition date, and picked the 5 latest deposited protein models (at the time of search) for every 10 validation score percentiles. I.e., we include 5 newest models with validation scores between 0 and 10%, 5 models with scores between 10 and 20%, and so on. The 5 RNA models were selected by sorting the models, which include RNA and have validation scores between 0 and 50%, by deposit date in PDBe. Some models from identical publications, as well as models associated with partial or unaligned CryoEM maps, were excluded from the process. From their original publication, the models were generated with both automatic refinement and manual adjustment, and the software used included Phenix, Coot, and NAMD[16,18,44]. The PDB

ID of the models are: 8t0q, 8gso, 8wyc, 8unh, 8xom, 8wx0, 8t0v, 8q74, 8xqp, 9c49, 8xzb, 9c57, 8k7t, 8xse, 8wly, 8xqr, 8k03, 8xqa, 8k3j, 8k11, 8ost, 9enf, 8tjv, 7yg9, 8uau.

For each of the 25 models, default GMM-based single model refinement was performed, using the PDB model and the corresponding CryoEM maps from EMDB as input. The geometry scores of both the original and refined models were calculated using MolProbity (which the PDB validation server is mostly based on) provided through Phenix; the RNA suite scores were computed using the SuiteName tool from Phenix; and the Q-score was calculated using the plugin in UCSF Chimera[26,29,38,40,45].

From the plot in Fig. 2E, the Q-scores of the models remain relatively constant after refinement, while the protein geometry score, as well as RNA scores, improve in every case Figs. 2E, and S5, 6. Note that better geometry is indicated by lower MolProbity and higher RNA suite score. This is expected since, unlike the TRPV1 example shown in Fig. 2A–D, in which we fit a model into a higher quality map of different conformation, here each model was built originally from the associated map. At a closer look, it is worth noting that for most of the entries, there is a small increase of Q-scores after the refinement, but for the few models with the highest initial Q-scores, our model refinement decreases the score slightly. This is because many sidechains in those models were tightly fitted into the map, without the concern of "allowed" rotamers. Since our refinement protocol finds the acceptable rotamers that fit best to the map, it led to much fewer rotamer outliers, but slightly worse sidechain fitting.

The exact time cost of the refinement depends on the size of the molecule, but for the examples of single model refinement shown in the paper, it takes a few minutes for one iteration, and less than an hour on a single GPU (Nvidia RTX A5000) for the full refinement. The time per iteration is not too different for the single or the model series refinement, but the later often takes more iterations to converge. The model series refinement of the spliceosome takes ~5 h on a GPU, and the refinement of the TRPV1 takes ~2 h.

## Resolution for model refinement

Conceptually, it is easy to assume models built from lower-resolution structures would have lower geometry scores. However, since we use FSC as the metric for map-model similarity, the model only needs to agree with the map at the target resolution, and the geometry of the model can be further refined without impacting the map-model FSC. Theoretically, assuming we have a perfect estimation of map resolution and model geometry, for a map determined at any resolution, there is at least one model that fits the map at the determined resolution, which also has the ideal geometry. For example, given a 5 Å CryoEM map, it should be possible to build a model with a perfect validation score that also agrees with the map. Without the high-resolution information, it is impossible to confirm the refined model is exactly the correct one, i.e., it will match the map precisely if the structure can later be determined at 3 Å or higher resolution. However, it is still a plausible model given the information from the map, as well as the prior information we have about protein geometry. So, it is safe to argue that such a model is clearly better than alternative models that also agree with the map at 5 Å resolution but with poor geometry scores. Additionally, since in many cases, the resolution of CryoEM structures is limited by the heterogeneity within the system, it is more likely that there are in fact many models with good geometry that agree with the map at 5 Å resolution, each of them being equally correct as each particle can adopt one of many conformations.

Using the GMM-based representation, we can refine the atomic coordinates in real space while evaluating the map-model similarity in the Fourier space. This makes it possible to explore the different models that match the 3D reconstructions equally well at the target resolution and find the one with the best possible geometry. Despite the capability, similar to all CryoEM practices, it is still risky to interpret the structures beyond that resolution, especially for small features such as the opening size of a protein channel. For single model refinement, since the resolution of the map is estimated by the gold-standard FSC curve, it is relatively easy to set that as the target resolution for the model refinement. The issue is more

complicated for the model series refinement, because there has not been an established method that estimates the resolution of the continuous movement trajectories resulting from heterogeneity analysis of CryoEM data. In the model series refinement examples presented in the paper, we decided the target resolution by visual inspection of the individual 3D reconstructions. This is likely an underestimation because the model refinement protocol gathers information from many reconstructions instead of an individual one, but it is still our current best practice without reliable resolution estimation for continuous movement.

### PDB validation metrics
In our refinement method, only metrics used in the PDB validation server are implemented to constrain the geometry of the molecules. Since many of the PDB validation metrics are based on statistics from decades-old literature with small sample sizes[26,40,46], it is debatable whether achieving a high validation score is necessary or sufficient for a good protein/RNA structure. For most of the validation metrics, the allowed std is much smaller than the resolution of even the best structures determined using CryoEM. For example, the std of most bond lengths is around 0.02 Å, so even an outlier would be only ~0.1 Å off from a structure with the ideal geometry. Very often, the difference between models with a good and poor score can be so tiny that they have virtually the same similarity score when compared to a CryoEM map at near-atomic resolution. While our method can produce molecular models with near-perfect validation scores, the actual quality of the models still depends on how accurate the stereochemical statistics are, and the model we generate can be biased if the validation metrics we use are not reliable.

To show the sensitivity of the validation metrics, in a more extreme example, we started from a model of apoferritin (PDB-8T4Q) with a perfect PDB validation score and refined it with an inverted geometry loss function. That is, the optimizer will search locally, starting from a good model, and look for a model with the worst possible geometry score. Surprisingly, a model with an extremely poor score at every metric can be produced, which has only 0.2 Å RMSD on average from the input good model, and no atoms shift more than 0.3 Å (Fig. S7). With such small movement, the two models would be virtually indistinguishable even when the map is determined at near-atomic resolution. As a result, for most CryoEM structures, the geometry scores of the molecular models are actually driven by the set of validation metrics, as well as the model refinement method used, instead of the actual information from the experimental maps.

Additionally, since PDB validation currently does not include scores for the geometry of DNA (other than bond length and angle) or many small-molecule ligands, the refinement of those features is only driven by the CryoEM maps. While improving the validation metrics will be an ongoing effort of the broad structure biology community, the DNN implementation of our method makes it relatively convenient to incorporate new, well-defined metrics as a part of the loss function during training, so the quality of output models can improve as the field moves forward.

### Statistics and reproducibility
The DNN based model refinement method is generally deterministic. While the weights in the neural networks are initialized with very small random values, the initialization has little effect on the result of refinement. With the same input map/model and same refinement parameter selection, the output models have no distinguishable differences.

### Reporting summary
Further information on research design is available in the Nature Portfolio Reporting Summary linked to this article.

### Data availability
All data used in the paper are publicly available through EMPIAR, EMDB and PDB. The PDB code for the refined structure of TRPV1 is 9OGK. The 25 refined PDB models in Table 1, as well as the model series depicting continuous movement of TRPV1 and Spliceosome, are deposited to Zenodo[47], with the https://doi.org/10.5281/zenodo.15319666.

### Code availability
All computational tools described here are implemented in EMAN2[48], a free and open source software for CryoEM/CryoET imaging processing. The code is available at github.com/cryoem/eman2, and a tutorial can be found through eman2.org/e2gmm_model. Following the policy of the journal, the exact version of EMAN2 at the time of publishing, can be accessed from Zenodo (https://doi.org/10.5281/zenodo.15319831)[49], but it is always recommended to install the latest version for any data processing task.

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

## Acknowledgements

This research is supported by NIH grant R01GM150905.

## Competing interests

The author declares no competing interests.
