## [Transparent Peer Review file · Communications Biology]

Building molecular model series from heterogeneous CryoEM structures using Gaussian mixture models and deep neural networks

Corresponding Author: Dr Muyuan Chen

Version 0:

Reviewer comments:

Reviewer #1

(Remarks to the Author)

The paper, "Building molecular model series from heterogeneous CryoEM structures using Gaussian mixture models and deep neural networks", presents a method that is able to produce series of molecular models from heterogeneous cryo-EM datasets, with good geometry score throughout, using trained neural networks.

The work appear promising, and the method seems adequate, but the experimental validation is limited, and make it difficult to gauge the robustness and applicability of the approach, beyond the example shown. More detailed comments below:

1) More rigorous statistical analysis of the behaviour of the model on a larger dataset would help demonstrate the method's strengths, with a comparison to existing reconstruction methods (comparisons with Phenix, Coot and NAMD are mentioned in the supplementary, but not shown).

2) Additionally, while far from being the only tool to do so, the direct use of Ramachandran restraints to prevent outlier is not a guarantee for quality: when a validation metric is used as the target, it loses its ability to correctly inform on the structure's quality, and may in fact reduce the model's quality, as pointed out by the MolProbity authors (Prisant et al., Protein Science 2020, 29 (1), 315–329. <https://doi.org/10.1002/pro.3786>). while this method is not the only one using such restraints, I think further evidence that the model's quality is improving would be warranted.

In the end, I want to add that I find the approach really interesting, and think it has merits; more robust validation and benchmarking of the method and results would improve the paper.

Reviewer #2

(Remarks to the Author)

This manuscript introduces a novel model refinement protocol that utilizes a Gaussian Mixture Model-Deep Neural Network (GMM-DNN) framework to optimize Fourier Ring Correlation with 2D maps while simultaneously fitting stereochemical constraints of molecular models. The authors validate the robustness of this method across multiple datasets, with its application to Continuous Model Series Refinement being particularly compelling. This work will attract significant interest from the structural biology community and related fields.

I recommend accepting this paper, provided the following issues are addressed:

1. The manuscript includes relatively few figures in the main text. It would benefit from restructuring to separate the results and discussion sections from the methods section. Furthermore, consider selecting figures from Figure S1, Figure S4, and Figure S5 as main text figures instead of supplementary figures to better illustrate the results.

2. The caption for Figure S5D appears incorrect. It does not include a blue model.

3. The authors emphasize using 2D projection images instead of 3D volumes for training. Is that consistent during the

inference stage? How do the authors ensure that the 2D projections used in inference accurately represent the 3D volume without missing information in some directions?

4. The process for integrating different levels of refinement needs further clarification. Additionally, I would like to know whether large-scale domain movements can be optimized using FRC alone, or whether the network incorporates alternative objectives, such as structural similarity to the deposited models.

5. I am curious about the computational efficiency of this protocol, particularly for Continuous Model Series Refinement. What is the computational cost of a single refinement iteration, and how does it impact the computational cost of continuous model series refinement? Does the protocol offer a time advantage over Phenix in single refinement?

6. Although this relates to the author's prior work, I am curious whether each frame in the continuous model series refinement corresponds to real-world temporal intervals or merely sampling in the latent space of the VAE.

Reviewer #3

(Remarks to the Author)

This manuscript describes a method to refine a molecular model against a single EM map or a series of EM maps of conformationally varying molecular complex like the EM maps reconstructed along different directions in a conformational landscape of the complex. The conformational landscape is a space that contains the conformational information of an ensemble of single particle images, where different single particle images are projected and represent different conformations of the complex. The manuscript describes a deep learning method that is based on representing the molecular structure with a Gaussian mixture model (GMM). The GMM and a deep learning network were used by the author in a previous publication that proposed a method focused on extracting the conformational heterogeneity from single particle images. In the present manuscript, the author proposes a method for atomic modeling of the EM maps obtained by the previous method or by other methods. Methods for this purpose exist already and are well adopted by the community (EM and modeling). The author shows a comparison of their method with real-space refinement in Phenix, but for only one example of refinement (a single model refinement of TRPV1, Fig. S5). It is not clear why the other EM maps used in the manuscript were not modeled using Phenix as well, especially the EM maps obtained from heterogeneous datasets. A systematic comparison of the method proposed in the manuscript with Phenix would be useful. It could help the users of Phenix to better see the advantages of using the complex new method proposed in the manuscript (based on 3 neural networks, each one specialized for a different task). Furthermore, the author seems to have overlooked the already published methods that allow a direct atomic modeling from each single particle image and obtaining series of atomic models from these images along different directions in the conformational space (e.g., MDSPACE, based on MD simulations, <https://doi.org/10.1016/j.jmb.2023.167951>). Similar methods exist for atomic modeling of sets of EM maps (e.g., MDTOMO for atomic modeling of subtomograms as well, <https://doi.org/10.1038/s41598-023-37037-9>). Why performing atomic modeling based on EM maps obtained by 3D reconstruction from conformationally heterogeneous particle images when solutions exist for atomic modeling based on individual 2D particle images? This should be discussed in the manuscript.

Additional points:

Line 364-365: "the input to the DNN is always a vector (4D by default) of constant ones"
Please clarify how this 4D vector is obtained and how it is used by the network.

Line 433-434: "the actual conformation value of the corresponding reconstruction is used as the input of the DNN"
Please clarify how the conformation value is used by the network and if a mentioned random value is added to this conformation value.

Lines 457-459: "The heterogeneous model series was built from a series of 3D reconstructions generated along the first eigenvector of the particle distribution in the conformation latent space"
What is the resolution of each of these 3D reconstructions? How many images were used for each reconstruction? Can you calculate half maps and FSC-based resolution using these half maps?

Lines 468-470: "A series of 3D reconstructions were generated along one eigenvector in the conformation latent space"
Which eigenvector? What is the resolution of each of these 3D reconstructions (same questions as in the previous point)?

Line 587-588: Figure S2, panel A. It is difficult to see the conformational differences in the series in panel A. Can you indicate the changes using arrows?

Line 612-613: Figure S5, panels C-D. There are some issues with colors and structures shown in panels C and D. If the view is the same in panels C and D, the original structure should be in the same position with respect to the cryo-EM map in these two panels. It seems that this is the pink structure, but the legend of panel C reads that the pink structure is the GMM-refined one and the legend of panel D does not mention the pink color. Also, I would expect that the original structure means the starting structure for the refinement and that this structure may be slightly outside of the EM map like the violet and cyan structures in panels C and D, respectively, but these structures are not in the same position with respect to the cryo-EM map.

Version 1:

Reviewer comments:

Reviewer #1

(Remarks to the Author)

The revision is welcome, but does not really address my concerns regarding the results, and the robustness of the method. More detailed comments below:

1. The addition of table 1 is welcome. It does show improvements in both Q-score and MolProbity on average: Q-scores averages are respectively 0.5128, 0.5148, 0.5208, for the original, GMM, and Phenix columns, and 2.22, 1.192, 2.02 for the MolProbity scores. I would suggest the author to include at least the means, although a statistical tests would be useful too. This values also run contrary to the author's statements that "[...] running another round of refinement with those software does not yield significant improvement." This highlight the need for further validation for the method. As it stands, given the provided data, it does not appear to reliably provide models of better fit than Phenix, and while the lower MolProbity scores may indicate higher-quality models, they may also be a sign of overfitting.

2. The total absence of Ramachandran outliers from the proposed method is concerning. Outliers are uncommon, but not inexistent, and this would hint, I think, at the presence of overly strong constraints. This explanation is consistent with the data shown in fig. 2E, where the points showing the greatest decrease in Q-score, also shown dramatic decrease in MolProbity scores.

3. A newly introduced statement regarding MD and bias in the text appear incorrect, as it stands: " It is worth noting that here we only use static constraints that describe the quality of output model, but not any of the force field from MD simulation. As such, the conformational changes captured by our modeling approach are only driven by the underlying CryoEM data." The presence of strong stereochemical constraints will have bias the conformational changes found by the method. Or, if they are not needed, and do not affect the conformational changes, this should be demonstrated, for example by providing paths with and without the stereochemical constraints.

The method still appear promising, and of interest to the community, provided it is accompanied by more systematic comparisons to other methods.

Reviewer #2

(Remarks to the Author)

The author has addressed all of my concerns and I look forward to more detailed code comments and documentation.

Reviewer #3

(Remarks to the Author)

The authors have revised their manuscript, but a few points should be revised further.

Regarding the comparison of MD-based approaches for analyzing cryo-EM data in terms of conformational variability landscapes and the "data-driven" approaches for which the proposed approach was built, the authors are emphasizing model bias and force field "constraints" as potential issues with MD-based approaches and are proposing that the model series built with their method can be used to compare with the results of MD-based approaches "as an unbiased validation for the heterogeneity analysis and the force field accuracy".

Regarding the model bias, the authors should remind that the conventional approach to the heterogeneity problem in the cryo-EM field, used in all cryo-EM data processing workflows, is still discrete classification (typically classification based on maximum likelihood) and it results in a few high-resolution conformational classes out of which atomic models can be derived. The MD-based approaches use such trusted (validated) atomic models, obtained from one cryo-EM dataset, for MD-based refinement using the same cryo-EM dataset (with all classes mixed). So, the point of these approaches is not to go back to cryo-EM data analysis after running neural-network-based ("data-driven") approaches for conformational variability analysis, but to go back to cryo-EM data analysis after running the classical, discrete-classification approaches. In this context, the mentioned model bias of MD-based approaches is equivalent to model bias of any flexible fitting method, even the one proposed in the manuscript. Although based on neural networks, the approach proposed in the manuscript uses an existing atomic model and is biased by this model.

Regarding the force fields used in the mentioned MD-based approaches, they are not used as constraints, but to move the model during the simulation, and result in physico-chemically valid models.

Regarding the validation of different approaches, the cryo-EM field is currently lacking tools for validating different approaches for obtaining conformational landscapes from cryo-EM data. Before such tools are developed and validated, the proposed approach for model refinement from EM maps obtained by one of such heterogeneity analysis methods cannot be used for an unbiased validation of the atomic models that can be obtained by MD-based approaches.

Besides, the results of the proposed approach will always depend on the resolution of the EM maps obtained from the "data-driven" approaches for conformational landscape determination while MD-based approaches obtain the atomic models directly from 2D images (without combining images into 3D reconstructions).

The manuscript should be revised to take into account these remarks. In particular:

Reformulate lines 46-50. Remove “, and can potentially introduce unwanted model bias during the analysis. To truly model the transition between conformational states captured from the CryoEM data,”

Reformulate lines 200-206. Remove “This avoids unwanted model bias from the MD constraints, and the resulting models from our method can be compared to MD simulation of the system, serving as a validation of the CryoEM heterogeneity analysis as well as the MD force fields.”

Minor points:

Line 395-396 “a vector (4D by default) of constant ones” is not clear and could be replaced by “a vector of ones of length 4 ([1,1,1,1])”

Line 461: “a constant vector of one” is not clear and could be replaced by “a vector of ones”.

Line 526: “one eigenvector” should be replaced by “the first eigenvector” if this is the case.

Version 2:

Reviewer comments:

Reviewer #3

(Remarks to the Author)

The author has addressed all of my concerns. I recommend publication of the manuscript.

Reviewers' comments:

Reviewer #1 (Remarks to the Author):

The paper, "Building molecular model series from heterogeneous CryoEM structures using Gaussian mixture models and deep neural networks", presents a method that is able to produce series of molecular models from heterogeneous cryo-EM datasets, with good geometry score throughout, using trained neural networks.

The work appear promising, and the method seems adequate, but the experimental validation is limited, and make it difficult to gauge the robustness and applicability of the approach, beyond the example shown. More detailed comments below:

1) More rigorous statistical analysis of the behaviour of the model on a larger dataset would help demonstrate the method's strengths, with a comparison to existing reconstruction methods (comparisons with Phenix, Coot and NAMD are mentioned in the supplementary, but not shown).

In current examples discussed in the paper (the 25 protein models in Figure 1E and the 5 RNA models in Figure S6), we compared our method to deposited models, which are already refined by the authors using Phenix, Coot and other software, according to the corresponding literature. So running another round of refinement with those software does not yield significant improvement. Additionally, because most deposited models also involve a fair amount of semi-manual refinement by the authors, re-refine it with existing fully automatic tools can sometimes undo those changes and make the result slightly worse. By running our refinement protocol on the final deposited models, we aim to show that our method can still improve on top of the best practice used by the authors in their papers. Note we select models from the more recent papers, so that the latest modeling software is used in the original publications. To further clarify, we now also run Phenix auto refinement on all examples to show the improvement, and include a list that compares the results (Table 1, Figure S5).

2) Additionally, while far from being the only tool to do so, the direct use of Ramachandran restraints to prevent outlier is not a guarantee for quality: when a validation metric is used as the target, it loses its ability to correctly inform on the structure's quality, and may in fact reduce the model's quality, as pointed out by the MolProbity authors (Prisant et al., Protein Science 2020, 29 (1), 315–329. <https://doi.org/10.1002/pro.3786>.) while this method is not the only one using such restraints, I think further evidence that the model's quality is improving would be warranted.

Indeed, ideally it would be good to use other constraints, as well as the information in the map, to refine the model, and save a few validation metrics to measure the quality of the model refinement. However, Ramachandran angles, which is defined as the combination of two dihedral angles, is very sensitive to the absolute position of individual atoms. As we have shown in Figure S7, a movement of $\sim 0.2\text{\AA}$ in atomic position can lead to a drastic change of Ramachandran score. So even with a CryoEM map at better than 3\AA resolution, it is still near

impossible to build a model with perfect Ramachandran angles using the information from the density map alone.

One alternative solution would be to have a more sophisticated way to optimize the hydrogen bond network and use that as a constraint for backbone geometry. However, it is challenging to reach the 0.2Å precision with this approach because hydrogen bonds have a wider range of possible length and angle. Additionally, it is difficult to have a full view of the hydrogen bond network without seeing all the water molecules, which still isn't possible for most CryoEM maps.

Another alternative solution is to have a deep neural network that learns the local backbone geometry from the PDB and use that to guide the refinement. While it might be easier to present it as a more modern, unbiased approach, it is more likely that the neural network simply learns to encode the Ramachandran plots in its weight matrices, and we are just introducing the same bias in a more hidden way.

In sum, it is difficult to get a good Ramachandran score without using itself as constraints during refinement. However, Ramachandran isn't the only one that faces the issue. For most other metrics, such as bond length, angles and planarity, we also use them both as constraints during refinement, and as validation items. It is worth noting that even when using the same metric for refinement and validation, it is always easy to get a perfect score. For example, the clash score is considered in virtually every model refinement method, but the DNN-based approach presented here produces fewer clashes than existing methods. This is likely because the complexity of the DNN is better suited to optimize the multi-body geometry of large complexes.

In the end, I want to add that I find the approach really interesting, and think it has merits; more robust validation and benchmarking of the method and results would improve the paper.

Reviewer #2 (Remarks to the Author):

This manuscript introduces a novel model refinement protocol that utilizes a Gaussian Mixture Model-Deep Neural Network (GMM-DNN) framework to optimize Fourier Ring Correlation with 2D maps while simultaneously fitting stereochemical constraints of molecular models. The authors validate the robustness of this method across multiple datasets, with its application to Continuous Model Series Refinement being particularly compelling. This work will attract significant interest from the structural biology community and related fields.

I recommend accepting this paper, provided the following issues are addressed:

1. The manuscript includes relatively few figures in the main text. It would benefit from restructuring to separate the results and discussion sections from the methods section. Furthermore, consider selecting figures from Figure S1, Figure S4, and Figure S5 as main text figures instead of supplementary figures to better illustrate the results.

The idea behind the current paper structure is to separate the content for software users and for people interested in the model refinement methods. So the main text is relatively short and focuses on displaying the results. Regardless, we have now moved Figure S1 to main figures. Figure S4 and S5 still seem a bit too detailed.

2. The caption for Figure S5D appears incorrect. It does not include a blue model.

Yes, the pink model is the GMM refined one in both C and D. When making the final figure, I decided to not overlay the input model with the Phenix refined version because the difference between them is too little. But I forgot to change the caption, causing confusion. It is now corrected.

3. The authors emphasize using 2D projection images instead of 3D volumes for training. Is that consistent during the inference stage? How do the authors ensure that the 2D projections used in inference accurately represent the 3D volume without missing information in some directions?

During the training process, we use 2D images to save GPU memory. Since the neural network goes through projections images covering all orientations many times, the missing information is not a major concern there. However, note that the images are only used to compute the loss function during training, not as input to the neural networks.

During the inference stage, no 2D projection or 3D volume is used as input. For the single model refinement, the input is always a constant vector of one for training and inference. The neural network actually captures the information of the model within its weight matrices, so during the inference, it is only necessary to input the same constant vector and get the refined model from output.

For the model series refinement, the neural network is trained that an input vector from the conformation space is mapped to a model at the corresponding conformation. So only a list of vectors are used at the inference stage. Again, the images are only used to compute loss functions during training, not as input, so they can simply be removed during inference.

4. The process for integrating different levels of refinement needs further clarification. Additionally, I would like to know whether large-scale domain movements can be optimized using FRC alone, or whether the network incorporates alternative objectives, such as structural similarity to the deposited models.

Yes, large scale domain movement can be optimized using FRC alone, without external information. This is best shown in our previous work that analyzes the structure flexibility from CryoEM particles using a similar framework. Fitting the model to a map with a different domain arrangement is essentially the same task as learning the domain motion from the particles, except the better SNR in the reconstructions makes the task easier. We now include a paragraph to emphasize the relationship to the previous work, and clarify the integration between the three levels of refinement.

5. I am curious about the computational efficiency of this protocol, particularly for Continuous Model Series Refinement. What is the computational cost of a single refinement iteration, and how does it impact the computational cost of continuous model series refinement? Does the protocol offer a time advantage over Phenix in single refinement?

The exact time cost depends on the size of the molecule, but it normally takes a few minutes for one iteration, and less than an hour for the full refinement. The time per iteration is not too different for the single vs continuous model refinement, but the later may take more iterations to converge. The examples shown in the paper take 2-5 hours of GPU time. The deep neural network training is more computationally complex than the real space refinement of Phenix, but the GPU computing makes it much more efficient. Still, a refinement run with this protocol is not faster than a Phenix refinement with the same parameters. However, the main time advantage we offer here is to achieve a model with good geometry in a single run, without the time spent for manually tweaking local clashes and running the refinement multiple times.

We now include a paragraph discussing the time cost of the process.

6. Although this relates to the author's prior work, I am curious whether each frame in the continuous model series refinement corresponds to real-world temporal intervals or merely sampling in the latent space of the VAE.

No. Since CryoEM only observes snapshots, there is not a straightforward way to connect the conformational variability from many particles to the real-world temporal intervals. It may become possible in the future, after more time-resolved structure studies characterize the speed of different conformational changes, which can be used as priors to map a VAE latent space to an actual conformation space with temporal meaning.

Reviewer #3 (Remarks to the Author):

This manuscript describes a method to refine a molecular model against a single EM map or a series of EM maps of conformationally varying molecular complex like the EM maps reconstructed along different directions in a conformational landscape of the complex. The conformational landscape is a space that contains the conformational information of an ensemble of single particle images, where different single particle images are projected and represent different conformations of the complex. The manuscript describes a deep learning method that is based on representing the molecular structure with a Gaussian mixture model (GMM). The GMM and a deep learning network were used by the author in a previous publication that proposed a method focused on extracting the conformational heterogeneity from single particle images. In the present manuscript, the author proposes a method for atomic modeling of the EM maps obtained by the previous method or by other methods. Methods for this purpose exist already and are well adopted by the community (EM and modeling).

The author shows a comparison of their method with real-space refinement in Phenix, but for only one example of refinement (a single model refinement of TRPV1, Fig. S5). It is not clear why the other EM maps used in the manuscript were not modeled using Phenix as well, especially the EM maps obtained from heterogeneous datasets. A systematic comparison of the method proposed in the manuscript with Phenix would be useful. It could help the users of Phenix to better see the advantages of using the complex new method proposed in the manuscript (based on 3 neural networks, each one specialized for a different task).

The TRPV1 shown in Figure 1 is actually a rare case. It is determined on a map with high enough resolution for full atom modeling, but is published early enough that the refinement and validation of models was not routine yet. So in this case we have a full atom model built solely from the map, prior to any refinement, and it is a good benchmark to test the refinement methods.

For the other cases (the 25 models in Figure 1E, the 5 models in Figure S6, and the spliceosome model for the heterogeneous analysis), they are already refined by the authors using Phenix and other software, often also with manual adjustment, according to the corresponding literature. So running another round of refinement with those same software does not significantly improve the results. To further clarify, we now also run Phenix auto refinement on all examples to show the improvement, and include a list that compares the results (Table 1, Figure S5).

Furthermore, the author seems to have overlooked the already published methods that allow a direct atomic modeling from each single particle image and obtaining series of atomic models from these images along different directions in the conformational space (e.g., MDSPACE, based on MD simulations, <https://doi.org/10.1016/j.jmb.2023.167951>). Similar methods exist for atomic modeling of sets of EM maps (e.g., MDTOMO for atomic modeling of subtomograms as well, <https://doi.org/10.1038/s41598-023-37037-9>). Why performing atomic modeling based on EM maps obtained by 3D reconstruction from conformationally heterogeneous particle images

when solutions exist for atomic modeling based on individual 2D particle images? This should be discussed in the manuscript.

These are two related, but different approaches to the structure variability problem. The model refinement method described in this paper is designed for a data driven approach of CryoEM data processing. That is, starting from a CryoEM dataset, one can achieve a high resolution 3D reconstruction of the molecule, and analyze the structure heterogeneity of the system just from the particles alone, without prior information. A model can then be built from the high resolution map, and the method described in this paper can be used to refine the model and build dynamic models for the conformational changes. Also, in this approach, we only include the stereochemical constraints to ensure the model at every frame has good geometry, but not any of the MD force fields. Therefore, any movement of the model produced here is driven directly by the heterogeneity of CryoEM data, and the model information does not impact the upstream particle-based analysis.

Alternatively, one can use the information from existing models from an early stage. This is indeed feasible, but it comes with a few potential issues. First, model bias can greatly affect the outcome. Using models for particle alignment can easily lead to overestimation of resolution, and it has been shown that using existing models for heterogeneity analysis can lead to incorrect results (Schwab et al. 2024). Second, MD simulation of large systems with large scale motion (such as the spliceosome example shown in Figure 3) based on 2D particles can be much more computationally expensive, especially if all atomic level constraints are maintained throughout the process.

Overall, whether to use the model information in the heterogeneity analysis stage is a rather philosophical question. I would argue that if it is possible to achieve similar results, it is ideal to use as little prior information as possible, so that any potential model bias is minimized. Since there are already multiple data-driven heterogeneity analysis tools, which can analyze the dynamics of the system from CryoEM data alone without the MD constraints and have already produced interesting results on realistic datasets, it would make more sense to have a method that build models on top of those results to better interpret them, instead of going back a step and re-analyze the heterogeneity from particles with MD assisted approaches. Additionally, since the force field is never used in the model building in our method, the model series built from heterogeneous CryoEM datasets can be used to compare with the MD simulation results and serves as an unbiased validation for the heterogeneity analysis and the force field accuracy.

We have now included the citations to the MD-based paper and some descriptions that highlight the difference of the different approaches.

Additional points:

Line 364-365: "the input to the DNN is always a vector (4D by default) of constant ones"
Please clarify how this 4D vector is obtained and how it is used by the network.

It is a constant vector, i.e. [1,1,1,1], and does not need to be obtained from anywhere. It is directly used as the input for the network.

Line 433-434: “the actual conformation value of the corresponding reconstruction is used as the input of the DNN”

Please clarify how the conformation value is used by the network and if a mentioned random value is added to this conformation value.

The conformation value is used as the input to the network, and a random value is added on top of it during training.

Lines 457-459: “The heterogeneous model series was built from a series of 3D reconstructions generated along the first eigenvector of the particle distribution in the conformation latent space”
What is the resolution of each of these 3D reconstructions? How many images were used for each reconstruction? Can you calculate half maps and FSC-based resolution using these half maps?

Measuring the resolution of a continuous conformation change from CryoEM is still an unsolved problem. Certainly it is possible to analyze the dynamics of the full dataset, produce two half maps of each frame along a continuous motion, and calculate the FSC between them. However, since the two half sets of particles have been mixed together during the analysis, those FSCs should not be considered “gold-standard”, and the resolution measured from them does not necessarily reflect the resolution of the structures.

Alternatively, it is also possible to keep the “gold-standard” particle split and analyze the heterogeneity in each half set independently. However, this leads to two independent latent spaces. While the general trend of movement should be the same from the two subsets, there is no obvious way to align the two latent spaces. So for one trajectory in the latent space from one half-set of particles, it is difficult to find the exact same trajectory in the latent space from the other half-set. Without the alignment, it is impossible to find the corresponding maps and calculate the “gold-standard” FSC for resolution measurement.

In sum, as mentioned in the “resolution for model refinement” section in methods, so far the most reliable way to estimate the resolution of a motion trajectory is still based on visual assessment of protein features. For example, we use 7Å for TRPV1 because the alpha helices are clearly visible in the reconstruction of frames, and use 15Å for the spliceosome because the RNA helices are visible.

Lines 468-470: “A series of 3D reconstructions were generated along one eigenvector in the conformation latent space”

Which eigenvector? What is the resolution of each of these 3D reconstructions (same questions as in the previous point)?

The first eigenvector from PCA. The maps are filtered to 15Å.

Line 587-588: Figure S2, panel A. It is difficult to see the conformational differences in the series in panel A. Can you indicate the changes using arrows?

A dashed line is now added to highlight the changes through the trajectory.

Line 612-613: Figure S5, panels C-D. There are some issues with colors and structures shown in panels C and D. If the view is the same in panels C and D, the original structure should be in the same position with respect to the cryo-EM map in these two panels. It seems that this is the pink structure, but the legend of panel C reads that the pink structure is the GMM-refined one and the legend of panel D does not mention the pink color. Also, I would expect that the original structure means the starting structure for the refinement and that this structure may be slightly outside of the EM map like the violet and cyan structures in panels C and D, respectively, but these structures are not in the same position with respect to the cryo-EM map.

Yes, the pink model is the GMM refined one in both C and D. When making the final figure, I decided to not overlay the input, unrefined model with the Phenix refined version because the difference between them is too little. But I forgot to change the caption, causing confusion. It is now corrected.

Reviewers' comments:

Reviewer #1 (Remarks to the Author):

The revision is welcome, but does not really address my concerns regarding the results, and the robustness of the method. More detailed comments below:

1. The addition of table 1 is welcome. It does show improvements in both Q-score and MolProbity on average: Q-scores averages are respectively 0.5128, 0.5148, 0.5208, for the original, GMM, and Phenix columns, and 2.22, 1.192, 2.02 for the MolProbity scores. I would suggest the author to include at least the means, although a statistical tests would be useful too. This values also run contrary to the author's statements that "[...] running another round of refinement with those software does not yield significant improvement." This highlight the need for further validation for the method. As it stands, given the provided data, it does not appear to reliably provide models of better fit than Phenix, and while the lower MolProbity scores may indicate higher-quality models, they may also be a sign of overfitting.

It is worth noting that the MolProbity score is a log-scaled summary of Rotamer, Ramachandran and clash score, and Q-score is a mostly linear description of the map-model fitting. For example, for a structure with the same Ramachandran score (PDB: 9enf), a clash score of 15 (i.e. 15 clashes per every 1000 atoms) results in a MolProbity score of 2.1, and a clash score of 1 leads to a MolProbity score of 1.0. On the other hand, a Q-score of 0.521 is equivalent to 3.31Å resolution, whereas a Q-score of 0.515 is equivalent to 3.34Å resolution. I believe it is obvious that a >90% reduction of clashing atoms is a major improvement in model geometry, and a change of map-model agreement of 0.03Å resolution is much less meaningful. To show the effect clearly, we now include the full score of Ramachandran, rotamer and clash score in the table, as well as the corresponding resolution of the Q-score.

The statement "running another round of refinement with (the same) software does not yield significant improvement" still holds true. First, a 0.03Å resolution improvement in map-model agreement does not appear significant. Second, in this comparison, we run another round of Phenix auto refinement on top of the structures deposited by the authors. The slight difference may come from the difference of software version, different refinement parameters, and some manual model editing by the authors before depositing their models. In general, model refinement is a deterministic process, and one should not expect different results by running the same software multiple times.

2. The total absence of Ramachandran outliers from the proposed method is concerning. Outliers are uncommon, but not inexistent, and this would hint, I think, at the presence of overly strong constraints. This explanation is consistent with the data shown in fig. 2E, where the points showing the greatest decrease in Q-score, also shown dramatic decrease in MolProbity scores.

Indeed, models with highest initial Q-scores tend to get the most improvement in geometry and decrease in map fitting. This is an indicator that those deposited models are overfitted to the density maps, and some geometry constraints are sacrificed in the process. Even in a case that the reviewer considered a “great decrease” (for example PDB:8wx0), where the GMM refinement results in a 0.05 decrease of Q-score, equivalent to 0.2Å change of resolution, but a 90% reduction of clashing atoms in the model. It is also worth noting the reported resolution of the corresponding CryoEM map, EMD-37896, is 3.7Å, is much lower than the resolution corresponding to the Q-score is 3.0-3.2Å, regardless of the refinement methods. So it is more likely the model is overfitted to the map (i.e. it is fitted at a higher resolution than the “gold-standard” resolution of the map) instead of having too strong constraints during model refinement.

The total absence of Ramachandran outliers is a different issue, but it is not necessarily an issue of any refinement approach. From the 25 models we tested, both methods produce >40% refined models with zero Ramachandran outliers. I agree statistically this is very unlikely to have no outlier at all among this many residues. However, I do not think this is caused by the overly strong constraints during the refinement. As shown in the paper (Fig S7), the percentage of Ramachandran outliers can change drastically with ~0.2Å RMSD of the model. The difference is much smaller than the pixel size of the map, and has virtually no effect on the map-model agreement with maps at >2Å resolution. Regardless how weak the Ramachandran constraints are set to be, as long as the refinement converges, the program can often easily find a solution with zero Ramachandran outlier and with equally good fitting to the map. Without a map at 1Å resolution, it would be difficult to tell whether a certain residue should be a real Ramachandran outlier directly from the data itself.

3. A newly introduced statement regarding MD and bias in the text appear incorrect, as it stands: " It is worth noting that here we only use static constraints that describe the quality of output model, but not any of the force field from MD simulation. As such, the conformational changes captured by our modeling approach are only driven by the underlying CryoEM data." The presence of strong stereochemical constraints will have bias the conformational changes found by the method. Or, if they are not needed, and do not affect the conformational changes, this should be demonstrated, for example by providing paths with and without the stereochemical constraints.

The point here is, when we retrieve a conformational change from CryoEM data, the data can only show the movement at 5Å or lower resolution. There would be infinitely many trajectories of molecular models at atomic level that agree with the movement derived from CryoEM data, and with our refinement protocol, we only find one of those trajectories with the best stereochemical geometry. Unlike some of the MD based methods, our model building process is not a part of the heterogeneity analysis, so obviously it does not introduce bias to the upstream analysis. However, when a researcher interprets the conformational changes based on the molecular models, it is certainly possible that their conclusion can be biased by the geometry constraints of the modeling process. We have now adjusted the text based on this and reviewer 3's comment.

The method still appear promising, and of interest to the community, provided it is accompanied by more systematic comparisons to other methods.

Reviewer #2 (Remarks to the Author):

The author has addressed all of my concerns and I look forward to more detailed code comments and documentation.

I will gradually add more code documentation to the Github repository. There is also the hope that some large language models can produce good documentation based on the code in the near future.

Reviewer #3 (Remarks to the Author):

The authors have revised their manuscript, but a few points should be revised further.

Regarding the comparison of MD-based approaches for analyzing cryo-EM data in terms of conformational variability landscapes and the “data-driven” approaches for which the proposed approach was built, the authors are emphasizing model bias and force field “constraints” as potential issues with MD-based approaches and are proposing that the model series built with their method can be used to compare with the results of MD-based approaches “as an unbiased validation for the heterogeneity analysis and the force field accuracy”.

Regarding the model bias, the authors should remind that the conventional approach to the heterogeneity problem in the cryo-EM field, used in all cryo-EM data processing workflows, is still discrete classification (typically classification based on maximum likelihood) and it results in a few high-resolution conformational classes out of which atomic models can be derived. The MD-based approaches use such trusted (validated) atomic models, obtained from one cryo-EM dataset, for MD-based refinement using the same cryo-EM dataset (with all classes mixed). So, the point of these approaches is not to go back to cryo-EM data analysis after running neural-network-based (“data-driven”) approaches for conformational variability analysis, but to go back to cryo-EM data analysis after running the classical, discrete-classification approaches. In this context, the mentioned model bias of MD-based approaches is equivalent to model bias of any flexible fitting method, even the one proposed in the manuscript. Although based on neural networks, the approach proposed in the manuscript uses an existing atomic model and is biased by this model.

I agree with the reviewer that in the case of discrete classification and model building afterwards, the MD-based methods introduce as much bias as any other modeling tools, including the one described here.

In this sentence, I meant to point out the risk of learning large-scale continuous conformational changes using MD-based constraints directly from the raw particle images (often in addition to machine learning techniques), which can lead to incorrect conclusions due to the high noise level in the images. Indeed, stating this in the introduction does appear too broad and can cause confusion. I have now removed the sentences as the reviewer requested.

Regarding the force fields used in the mentioned MD-based approaches, they are not used as constraints, but to move the model during the simulation, and result in physico-chemically valid models.

Regarding the validation of different approaches, the cryo-EM field is currently lacking tools for validating different approaches for obtaining conformational landscapes from cryo-EM data. Before such tools are developed and validated, the proposed approach for model refinement from EM maps obtained by one of such heterogeneity analysis methods cannot be used for an unbiased validation of the atomic models that can be obtained by MD-based approaches.

Besides, the results of the proposed approach will always depend on the resolution of the EM maps obtained from the “data-driven” approaches for conformational landscape determination while MD-based approaches obtain the atomic models directly from 2D images (without combining images into 3D reconstructions).

Yes, at the current point, it is arguable whether the conformational change retrieved from CryoEM data and the MD simulation can cross-validate each other. There are individual cases, for example tilting of alpha helices, where the conformational change can be validated by the basic geometry of the secondary structures, and resolution of maps can be estimated from the CryoEM reconstructions. But generally, I agree the validation is not widely applicable.

The manuscript should be revised to take into account these remarks. In particular:

Reformulate lines 46-50. Remove “, and can potentially introduce unwanted model bias during the analysis. To truly model the transition between conformational states captured from the CryoEM data,”

Reformulate lines 200-206. Remove “This avoids unwanted model bias from the MD constraints, and the resulting models from our method can be compared to MD simulation of the system, serving as a validation of the CryoEM heterogeneity analysis as well as the MD force fields.”

Minor points:

Line 395-396 “a vector (4D by default) of constant ones” is not clear and could be replaced by “a vector of ones of length 4 ([1,1,1,1])”

Line 461: “a constant vector of one” is not clear and could be replaced by “a vector of ones”.

Line 526: “one eigenvector” should be replaced by “the first eigenvector” if this is the case.

I have made the changes as the reviewer requested.